# Mindfulness Stress Management for Female Cancer Survivors Facing the Uncertainty of Disease Progression: A Randomized Controlled Study

**DOI:** 10.3390/ijerph19084497

**Published:** 2022-04-08

**Authors:** Jih-Teng Lee, Yi-Hua Lee, Yuan-Ping Chang

**Affiliations:** 1Department of Surgery, Fooyin University Hospital, Kaohsiung 928005, Taiwan; scott0813@gmail.com; 2Department of Administration, National Health Research Institutes, Zhunan 350401, Taiwan; yr7275@nhri.edu.tw; 3Department of Nursing, Fooyin University, Kaohsiung 831301, Taiwan

**Keywords:** female cancer survivors, mindfulness stress management, fear of recurrence, quality of life

## Abstract

This study aimed to determine the effect of a mindfulness stress management intervention on fear of recurrence and quality of life among female cancer survivors. A longitudinal, randomized design with two groups (60 participants) was used for pretest/posttest comparisons. Twelve weeks of mindfulness stress intervention effectively attenuated fear of recurrence symptoms (T1 *p* = 0.002, T2 *p* = 0.047), and quality of life (T1 *p* = 0.000, T2 *p* = 0.001) significantly increased. The results were significantly different between the intervention and control groups. Over time, group differences became more significant (T1 *p* = 0.002), demonstrating the effectiveness of the mindfulness stress management intervention. When female cancer survivors face uncertain disease progression, fear of recurrence affects their quality of life. When these women receive supportive intervention sooner, their improvement is more significant. Healthcare providers should encourage female cancer survivors to engage in mindfulness stress management actions to achieve a better benefit.

## 1. Introduction

In Eastern culture and society, women generally perform a significant role in the family. Family dysfunction thus usually occurs when women suffer from cancer. Social and psychological support are essential needs [1]. In the sequence of illness and treatment, female cancer survivors usually experience psychological and physical symptoms of stress responses and exhibit cancer-associated mood disorder, anger, confusion, fatigue, anxiety, depression, insomnia [2,3,4], and traumatic life experiences [5]. Compared to healthy women, female cancer survivors most commonly experience mood disorders caused by fear of recurrence (FOR), such as a high level of psychological insecurity, anxiety, and depression [6]. Therefore, supportive care and lifestyle rehabilitation for female cancer survivors should be given specific attention. Patients with these mood disorders can improve their quality of life (QOL) and their physical/psychological health through supportive interventions to achieve effective emotional management [6,7].

Due to the uncertainty of prognosis and disease progression following cancer treatment, cancer recurrence is the most difficult stage in the illness trajectory, and recurrence is the situation most feared by cancer survivors when facing uncertainty regarding their life and health. Because cancer survivors bear higher stress, anxiety, depression, and physiological symptoms than normal individuals, they constitute a group that requires specific psychological intervention support [8,9]. Compared to male cancer survivors, female cancer survivors have a higher degree of FOR, which severely affects their QOL and burdens the patients themselves, their families, and society [6,7].

The mindfulness stress management (MSM) intervention adopted in this study is based on mindfulness meditation-based stress reduction used in cancer patients. MSM provide self-awareness and concentration training through group participation and daily homework to nonjudgmentally observe and accept a difficult reality. Mindfulness involves actively adjusting negative thinking and responses to guide better response strategies, promote positive personality development, avoid negative thinking, correct aberrant cognitive behaviours and counter low moods caused by FOR to improve physical and physiological health [2,7,8,10]. Mindfulness training can also be used to promote resilience and physical and mental adaptability [11], to instil a more optimistic attitude towards life, to improve interpersonal relationships, and even to face death with more equanimity [12]. However, malnutrition is widespread among cancer survivors; proper nutrition education can help them to improve their QOL [13], and complete nutritional supplementation can reduce the physical and mental stress caused by cancer [14].

The most common cancers in women include breast cancer, cervical cancer, uterine corpus cancer, and ovarian/fallopian tube cancer. According to global cancer statistics from the World Health Organization (WHO), there were 19.3 million new cases of cancer and almost 10 million cancer deaths in 2020. Among the top 10 cancer incidence rankings among women, breast cancer (24.5%), cervical cancer (6.5%), uterine corpus cancer (4.5%), and ovarian cancer (3.4%) are ranked first, fourth, sixth, and eighth, respectively. Among the top 10 female cancer mortality rankings, breast cancer (15.5%), cervical cancer (7.7%), and ovarian cancer (4.7%) are ranked first, fourth, and eighth, respectively [15]. In particular, breast cancer ranks first in women’s cancer worldwide and in Taiwan. Furthermore, the incidence of breast cancer in younger women has increased in recent years. A continued increase in morbidity is also predicted in the future. Therefore, follow-up treatment and supportive care issues are worthy of attention.

This study used a 12-week psychological intervention MSM support course through a randomized controlled trial to investigate the effectiveness of this intervention in addressing FOR and QOL in female cancer survivors. This study assessed the efficacy of health care providers as an important component of conventional cancer treatment care in the future.

## 2. Methods

### 2.1. Study Design and Participants

This study was a randomized controlled trial that used a longitudinal design with two groups measured by a pretest and posttest. The inclusion criteria were females aged 20–75 years with cancer who had undergone conventional cancer treatment for at least one month. Female cancer survivors who had been diagnosed with stage IV cancer and confirmed cancer recurrence were excluded. G*Power 3.1 was used to determine the sample size with an F test (ANOVA: repeated measure, within–between interaction). With an effect size = 0.25, alpha = 0.05, and power = 0.95, 54 participants were required for this study. A total of 60 participants were referred by outpatient physicians according to the protocol. Then, according to the order of referral, the odd number participants were assigned to the MSM intervention group (*n* = 30), and the even number participants were assigned to the control group (*n* = 30). Before the MSM intervention course, the participants in both groups received a two-hour nutritional education course, which served as the basic standard condition of the two groups. Then, the intervention group received MSM intervention for two hours every week for a total of 12 weeks. Both groups were administered the FOR and QOL questionnaire surveys three times (before the implementation of the MSM intervention (pretest, T0), after 12 weeks of the MSM intervention (posttest, T1) and at the three-month follow-up (posttest, T2)) to evaluate the degree of improvement and the effectiveness of the intervention. However, only 54 participants completed the intervention and all three measurements. Twenty-nine of these were in the MSM group (*n* = 29) and 25 were in the control group (*n* = 25) (Figure 1).

### 2.2. Intervention Programs

The basic course on nutrition education was guided by professional dietitians in two groups and emphasized the importance of a balanced diet and nutritional supplements to change previous poor eating habits. The MSM course for the intervention group was led by a senior clinical counselling psychologist, who was one of the scholars involved in the development of a mindfulness stress reduction program in Taiwan. The program was led according to the mindfulness meditation-based stress reduction skills framework. In addition, group counselling was provided to facilitate group member sharing and physical/mind relaxation training. A two-hour MSM group intervention was held once per week for a total of 12 weeks. The first 90 min of the course each week featured group member sharing and guidance by the group leader to improve self-awareness and concentration. Through weekly member gatherings and homework practice, the participants developed a non-judgemental awareness of their thinking and emotions to help achieve a sense of inner peace and psychological balance. In addition, through group member sharing and guidance by the group leader, the participants developed strategies to respond to cancer-related personal issues. The last 30 min of each member gathering emphasized mindfulness-based stress reduction training and meditation through breathing exercises and body relaxation. This process consisted of guided breathing practices and training in body awareness and mood relaxation to achieve the effect of mind transformation and stress release. After each group activity, instructional CDs containing the exercises were provided so that the participants could practice at home every day to help further improve their body and mind relaxation skills and to remind them to practice this experience in their daily lives.

### 2.3. Measurements

The questionnaires included personal basic demographic and related disease information, and the FOR and QOL questionnaires. It took approximately 20–25 min to complete the two questionnaires.

FOR questionnaire:

The Chinese version of the FOR questionnaire was verified by Lee et al. (2018) [16] based on the Fear of Cancer Recurrence Inventory (FCRI) developed by Simard and Savard [17], with a total of 42 items. This measurement scale used a five-point Likert scale. The degree of FOR in the past several months was measured. Participants who had higher scores had higher FOR symptoms. The exploratory factor analysis used the Kaiser–Meyer–Olkin (KMO) measure (KMO value = 0.915) and Bartlett’s test of sphericity (*p* < 0.001). Fifteen items were retained and were divided into three domains: psychological distress, lifestyle function, and triggers; the Cronbach’s α for each dimension was 0.956, 0.936, and 0.794, respectively. The overall Cronbach’s α of the internal consistency reliability of the scale was 0.954, indicating that it was a reliable and practical measurement tool for assessing FOR.

QOL questionnaire:

QOL was measured using the fourth Chinese version of the Functional Assessment of Cancer Therapy-General (FACT-G), authorized by the Assessment of Chronic Illness Therapy (FACIT) Institute, which was developed by Cella et al. (1993) [18], with a total of 27 items. This measurement scale uses a five-point Likert scale to measure QOL in the past week. This questionnaire was divided into four domains: physical well-being and emotional well-being dimensions, where the higher the score is, the worse the QOL, and for the social/family well-being and functional well-being dimensions, the higher the score is, the better the QOL. The Cronbach’s α for the four subscales ranged between 0.81 and 0.93, and test–retest reliability ranged from 0.74 to 0.85, indicating excellent internal consistency [19].

### 2.4. Data Analyses

All participants completed the questionnaire surveys three times using a computer to key in the results. IBM^®^ SPSS Statistics 22.0 software was used for the descriptive and inferential statistical analyses. The distribution homogeneity of basic information for the two groups was examined using chi-squared testing. The effect of the intervention over time was analysed using paired sample t tests. Analysis of covariance (ANCOVA) was performed to analyse whether the improvement effect significantly differed between the intervention and control groups.

### 2.5. Ethics Approval

This study was approved by the Institutional Review Board (Approval No. FYH-IRB-108-10-01). Informed consent was obtained from all individual participants included in the study.

## 3. Results

A total of 54 valid participants who met the criteria completed this study. All participants completed the questionnaire three times. According to the statistical results of the pretest data (T0), most of the participants (46 patients, 85.2%) had breast cancer. There were only eight cases (14.8%) of gynaecological cancers. Twenty-two participants (40.7%) were aged between 50 and 59 years. Stage II cancer was the predominant stage type (21 cases, 38.9%). Approximately two-thirds of the participants survived more than five years. Approximately one-third of the participants were aware of symptoms of anxiety and depression. However, the pretest results of the FOR questionnaire showed that 32 people (59.3%) had mild degrees of FOR, 16 people (29.6%) had moderate degrees of FOR, and five people (9.3%) had severe degrees of FOR. After the random distribution into groups, the distribution of demographic information between the intervention and control groups did not significantly differ (*p* > 0.05), indicating that the samples in these two groups were homogeneous (Table 1).

Based on the paired sampled t test results from the two posttests, the intervention group significantly improved in FOR over time in both posttests. In addition, the level of attenuation of mental disturbance symptoms in the posttest following 12 weeks (T1 t = −3.380, *p* = 0.002, 95% CI = −0.579 to −0.142) was more evident than in the three-month follow-up posttest of the MSM intervention (T2 t = −2.081, *p* = 0.047, 95% CI = −0.510 to −0.004). The control group, which received only nutrition education, significantly experienced sustained increases in the severity of FOR symptoms in the posttest following the 12-week treatment (T1 t = 2.122, *p* = 0.044, 95% CI = 0.008 to 0.625), indicating that FOR was always present and did not disappear over time. The statistical results show that the improvement in QOL in the two groups significantly increased over time in both posttests. In addition, the level of QOL improvement was more evident in the MSM group (T1 t = 4.186, *p* < 0.001, 95% CI = 0.255 to 0.744; T2 t = 3.934, *p* = 0.001, 95% CI = 0.228 to 0.727) than in the control group (T1 t = 3.116, *p* = 0.005, 95% CI = 0.125 to 0.616; T2 t = 3.079, *p* = 0.005, 95% CI = 0.124 to 0.629) (Table 2).

In the pretest (T0), the average value and standard deviation of the FOR score for the intervention group (2.560 ± 0.861) were higher than those for the control group (2.237 ± 0.698). This result might not accurately depict the difference in the improvement effect between these two groups. Therefore, the confounding factor of the pretest value (T0) was controlled, and the corrected pretest value was used as a predictive variable. The premise of implementing ANCOVA is the assumption of the homogeneity of regression slopes. Once the *p* values pertaining to the group covariate interactions were confirmed to be greater than 0.05 (thus meeting the basic assumption), ANCOVA could be applied. Levene’s test was used to analyse whether the random samples in these two groups were homogeneous. The results meet the basic assumption of equal variances (*p* > 0.05). The final statistical results show that after the influence of the T0 data in these two groups was controlled for, the differences in the corrected means of the T1 were significant (*p* = 0.002) (Table 3).

## 4. Discussion

Official statistical data from Taiwan and elsewhere indicate that breast cancer has the highest incidence rate among cancers in women, which was consistent with the sample in this study. The incidence of breast cancer and gynaecological cancer in Taiwan has continued to increase in recent years, and the average patient age is decreasing. In addition to maintaining normal family functions and worrying about the uncertainty of disease progression, there are additional life challenges that these patients must face. These cancer survivors are a high-risk population for FOR. The level of psychological disturbance of cancer survivors will gradually increase with disease progression uncertainty, the occurrence of new health problems, and increased survival time [20]. In particular, the degree of FOR is more pronounced and severe in cancer patients with lower socioeconomic status and in younger female cancer patients [6,21]. Therefore, these patients particularly need psychological interventional support. The results from this study show that all participants had varying degrees of FOR. However, the participants were unaware of their FOR. More than 75% of the participants thought that they did not have symptoms of anxiety or depression. Nonetheless, after evaluation with the FOR questionnaire, more than half of the participants clearly experienced mild physical and mental disorders associated with FOR, and more than one-third of the participants experienced moderate to severe physical and mental disorders associated with FOR. This population severely lacked psychological support care services despite their urgent need for them. These results are consistent with findings from related research [6].

The findings of this study show that the FOR symptoms in the intervention group were effectively attenuated and that QOL significantly increased after MSM intervention, and the difference reached statistical significance. The improvement in the intervention group was greater than that in the control group. Group counselling was carried out to teach MSM skills, positive response modes and response strategies that help with emotional adjustments and recovery from negative emotions as early as possible. As a result, when a cancer survivor encounters disease progression uncertainty or daily life stressors, such strategies can reduce negative thinking, improve interpersonal interactions and social relationships, and increase the ability of cancer survivors to adapt psychologically and physiologically. Through group member sharing and MSM, FOR and negative thinking can be effectively reduced, and anxiety and depression can be mitigated to further increase QOL. These findings correspond to the results of other relevant studies [2,6,7,8,10,21,22,23].

Women with cancer experience more severe pressure and mood disturbances than healthy women. Their endocrine and immune system regulatory functions suffer long-term damage, producing adverse effects on disease prognosis, QOL, emotional function, and survival rates [2,22]. Because the subjects of this study were females with cancer, their FOR symptoms were even more severe than those of male cancer survivors [6,7,21]. Their physical functions and mood were severely affected for a long period, and their QOL was poor. Therefore, it is necessary to provide these patients with persistent interventional support. The FOR symptoms in the control group worsened, but their QOL significantly increased. One of the factors for this QOL improvement might have been the intake of a more balanced diet after attending the dietary training course. Good nutrition is an essential part of cancer treatment [13]. Sullivan et al. (2021) [24] indicated that most cancer survivors report serious nutritional problems after cancer, lack correct dietary information, and hope to obtain more nutritional support to improve potential nutritional-related problems. Karim et al. (2021) [25] indicated that nutritional guidelines for cancer are associated with reduced cancer incidence and mortality, with breast cancer having the most considerable reduction, followed by endometrial cancer. Therefore, nutrition education is also an indispensable and important step in cancer education, although the FOR symptoms from this study showed that nutrition education alone did not have a significant effect. This study showed that approximately 60% of the participants had mild FOR symptoms, and >65% of the participants had survived for more than five years. Through psychological interventional support, the disturbance caused by FOR was significantly attenuated, resulting in a steady trend of improvement. Therefore, patients should be provided with psychological interventional support as early as possible, preferably before anxiety or depression is aggravated, to more effectively improve their physical and mental health.

### Limitations

This study still has research limitations. However, we distributed the instructional CDs of MSM courses and dietary cancer guidelines to the two groups of participants so that they could continue to practice at home. However, during the three-month follow-up period, a regular follow-up mechanism was not implemented to understand the practice conditions of the new lifestyle in daily life. Therefore, the long-term improvements in physical and mental health well-being may be compromised due to a lack of consistent practice or insufficient hours of training. In addition, there is no particular emphasis on stress-relieving diets in the two-hour nutrition session education, which is our future research direction.

## 5. Conclusions

When stressful events and situations persist over time, both psychological and physiological health are affected. Cancer is a severely stressful situation for patients. The QOL of patients can be affected due to FOR as they face the uncertainty of cancer progression. When patients receive psychological interventional support earlier, the stress response effect may be better. Nutritional adjustment is the basis for physical recovery after cancer and plays an important role in cancer patients’ well-being. Furthermore, implementing nutritional improvements in daily life is easy. Medical care providers should encourage female cancer survivors to improve their body, mind, and nutrition together to obtain the best results. Based on the results of this study, MSM could indeed effectively help female cancer survivors to restore their physical and mental health, attenuate their anxiety, depression, or mood disorders, and increase their QOL. We expect MSM and nutrition education to be incorporated into routine cancer therapy simultaneously to help cancer survivors to maintain an optimal physical and mental status while facing uncertainty regarding disease progression to support their roles and functions within their families and society.

## Figures and Tables

**Figure 1 ijerph-19-04497-f001:**
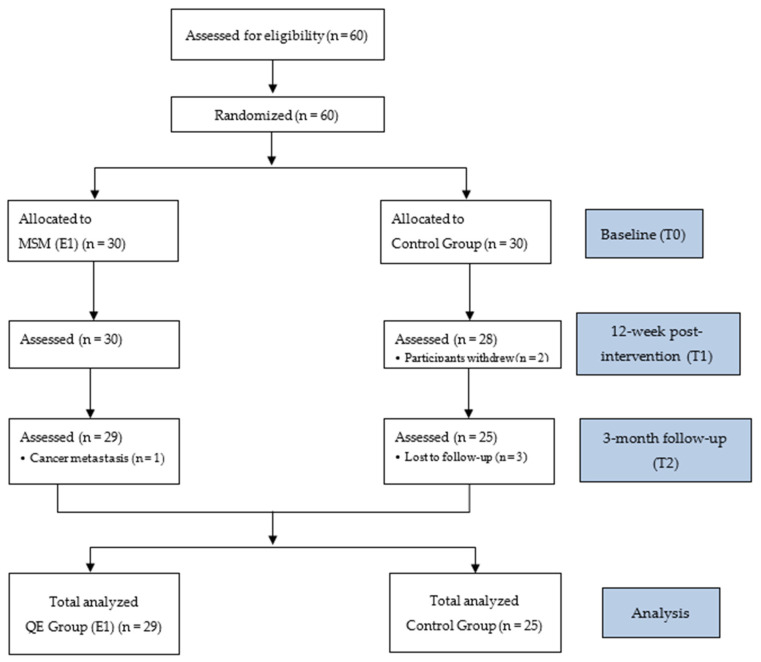
CONSORT flow diagram of this study.

**Table 1 ijerph-19-04497-t001:** Baseline characteristics of participants (*n* = 54).

Variable	Frequency Distribution (%)						Chi-Squared Test for Homogeneity	
	MSM (*n* = 29)		Control (*n* = 25)		Total (*n* = 54)			
	*n*	%	*n*	%	*n*	%	χ^2^-Value	*p*-Value
Age (y/o)								
30–39	1	3.4	2	8.0	3	5.6	3.505	0.320
40–49	7	24.1	7	28.0	14	25.9		
50–59	10	34.5	12	48.0	22	40.7		
≥60	11	37.9	4	16.0	15	27.8		
Marital status								
Single	4	13.8	5	20.0	9	16.7	1.107	0.893
Married	17	58.6	12	48.0	29	5.37		
Separation	2	6.9	2	8.0	4	7.4		
Divorced	4	13.8	5	20.0	9	16.7		
Widowed	2	6.9	1	4.0	3	5.6		
Education								
Below high school	0	0.0	6	24.0	6	11.1		
High school	11	37.9	10	40.0	21	38.9		
College and higher	18	62.1	9	360	27	50.0		
Cancer type								
Uterine cancer	3	10.3	2	8.0	5	9.2	0.460	0.928
Ovarian cancer	2	6.7	1	4.0	3	5.6		
Breast cancer	24	82.8	22	88.0	46	85.2		
Cancer stage								
I	12	41.4	8	32.0	20	37.0	3.064	0.382
II	9	31.0	12	48.0	21	38.9		
III	8	27.6	5	20.0	13	24.1		
Survival period								
Within 5 years	9	31.0	9	36.0	18	33.3	3.477	0.176
5–10 years	12	41.4	14	56.0	26	48.1		
>10 years	8	27.6	2	8.0	10	18.5		
Sense of anxiety								
No	21	72.4	15	60.0	36	66.7	0.931	0.335
Yes	8	37.6	10	40.0	18	33.3		
Sense of depression								
No	23	79.3	16	64.0	39	72.2	1.569	0.210
Yes	6	20.7	9	36.0	15	27.8		
Level of FOR Scale								
None	0	0.0	1	4.0	1	1.9	1.914	0.590
Mild	16	55.2	16	64.0	32	59.3		
Moderate	10	34.5	6	24.0	16	29.6		
Severe	3	10.3	2	8.0	5	9.3		

MSM, mindfulness stress management; FOR, fear of recurrence. A *p* value > 0.05 indicates that the distributions of the two groups are the same.

**Table 2 ijerph-19-04497-t002:** Paired sample t tests of outcome variables over time for within-group analysis.

Variables	MSM Group (*n* = 29)				Control Group (*n* = 25)			
	Mean	SD	t Value	*p* Value	Mean	SD	t Value	*p* Value
FOR								
T0	2.560	0.861			2.237	0.698		
T1	2.200	0.855			2.554	0.768		
T2	2.303	0.773			2.368	0.806		
T1–T0	−0.361	0.575	−3.380 **	0.002	0.317	0.747	2.122 *	0.044
T2–T0	−0.257	0.666	−2.081 *	0.047	0.130	0.705	0.927	0.363
QOL								
T0	3.546	0.610			3.621	0.832		
T1	4.046	0.588			3.992	0.554		
T2	4.049	0.598			3.998	0.600		
T1–T0	0.500	0.643	4.186 ***	0.000	0.371	0.595	3.116 **	0.005
T2–T0	0.478	0.642	3.934 **	0.001	0.377	0.612	3.079 **	0.005

MSM, mindfulness stress management; SD, standard deviation; FOR, fear of recurrence; QOL, quality of life. T0, pretest (baseline); T1, 12-week posttest; T2, three-month follow-up test. * *p* value < 0.05; ** *p* value < 0.01; *** *p* value < 0.001 (two-tailed).

**Table 3 ijerph-19-04497-t003:** The ANCOVA of outcome variables over time in MSM versus control group (*n* = 54).

Outcome Variables		The Assumption of Homogeneity of Regression Slopes		Levene’s Test of Equality of Error Variances		Tests of Between-Group Effects		Adjusted Posttest Means	
		F Value	*p* Value *^§^*	F Value	*p* Value *^§^*	F Value	*p* Value	MSM	Control
FOR	T1	1.113	0.296	1.995	0.164	11.209	0.002 **	2.097 ^†^	2.674 ^†^
	T2	0.053	0.819	0.619	0.435	2.357	0.131	2.210 ^†^	2.476 ^†^
QOL	T1	0.089	0.767	3.112	0.084	0.446	0.507	4.062 ^‡^	3.974 ^‡^
	T2	0.101	0.752	2.576	0.115	0.290	0.593	4.060 ^‡^	3.986 ^‡^

MSM, mindfulness stress management; FOR, fear of recurrence; QOL, quality of life. T1, 12-week posttest; T2, three-month follow-up. *^§^*
*p* value > 0.05 means the homogeneity assumption of regression slopes and variances in the two groups are both met. ^†^ Adjusted mean of covariate score: FOR baseline (T0) = 2.411. ^‡^ Adjusted mean of covariate score: QOL baseline (T0) = 3.581. ** *p* value < 0.01 (two-tailed).

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
