# Peer review of "Mindfulness Stress Management for Female Cancer Survivors Facing the Uncertainty of Disease Progression: A Randomized Controlled Study"

_ijerph, 2022, doi:10.3390/ijerph19084497_

Round 1

Reviewer 1 Report

The Authors present a paper" Mindfulness stress management for women cancer survivors facing the uncertainty of disease progression: a randomized controlled study" very well written, interesting and understandable for readers . Introduction , Methodology, Statistical analysis, Results and Discussion followed by Conclusions are clear and reporting practical suggestions to overcome the described problem.

I have only 2 minor requests:

  1. the mindfulness training should be led by adequately prepared  senior clinical counseling? do they need specific training courses?
  2. please indicate the median time to fill the questionnaires by cancer survivors

Author Response

  • Author's Response 1

 Yes. The mindfulness mentor has a Ph.D. education qualification, a mindfulness training certificate, and many years of counselling experience. He is a professor at the university.   

The mentor's qualification for the mindfulness course was added as follows:

The MSM course for the intervention group was led by a senior clinical counselling psychologist, who was one of the scholars involved in the development of a mindfulness stress reduction program in Taiwan.

  • Author's Response 2

We add the information on page 4 of 10 (Section 2.3)

The questionnaires included personal basic demographic and related disease information, and the FOR and QOL questionnaires. It took approximately 20-25 minutes to complete the two questionnaires.

Reviewer 2 Report

Introduction.

The first sentence seems strange to me. Both parents play an important role, it cannot be weighed or measured, and it is not related to the authors' research. Family dysfunction is also not the aim of the study, but the fear of relapse and the associated quality of life. The authors need to rethink the beginning of this section. This misleads the reader. The authors refer to the source [4] as being related to cancer, but this article is not about cancer, but about stress, mindfulness, cognition. The same goes for the source [6], which does not provide a comparison of treatment, as indicated by the authors. It is recommended to write the statistics not at the end of the chapter, but at the beginning.

Methods.

Some exclusion criteria has been mentioned, but inclusion criteria has not been mentioned, it also has not been mentioned how the selection was formed. It is not clear whether the selection was indeed randomized, as the impression from the text is that the available selection was used, which was then randomized, but then this is no longer a randomized controlled trial. In addition, it is not clear how the list of respondents was compiled.

FOR questionaire – it was not mentioned whether this instrument was adapted in this particular country, who did it (no reference) and what are the internal consistency indicators. If the survey has not been standardized in the country, then they cannot write “Participants who had higher scores had higher FOR symptoms”, respectively, this cannot be divided into grades. This needs to be clarified so that the situation is clear. In addition, in their part of the method, the authors describe the data of the study of the authors of the original survey [15] (1704 participants; Cronbach's alpha = 0.95), which is not relevant for this study.

The same goes for QOL. It is not clear whether the instrument has been validated, who did it, and so on. The authors cite the reference [16] as well as the data from its study, emphasizing internal consistency, but do not indicate whether they have calculated this indicator themselves.

Discussion

Both in this section and in the methods section, the authors mention diet education, but the introduction part does not say what it means for these patients. Also the control group’s QOL  indicator improvement is also associated only with diet education and not with other factors such as the absence of relapse over time and better well-being.

Limitation

One-sided description, emphasizing again the short time of diet education, but it has not been highlighted at all that the respondents fill in self-assessment questionnaires, that they can provide socially desirable answers, that the number of respondents is not large, that the available selection is chosen, etc.

Conclusion

The authors really emphasize a lot of nutritional improvements when it comes to physical health, but they do not talk about physical activities that are important in any person's life, they do not talk about body changes and their perception, especially women who have undergone mastectomy, the possibility of corrective surgery, etc.

Author Response

  • Author's Response 1

Some sentences have been deleted, and paragraphs have been adjusted.

Reference [4] and [6] have been deleted.

4. Garland EL (2007) The meaning of mindfulness: A second-order cybernetics of stress, metacognition, and coping. Complement Health Pract Rev 12(1):15–30. http://dx.doi.org/10.1177/1533210107301740

6. Hsieh CC, Chen CA, Hsiao FH, Shun SC (2014) The correlations of sexual activity, sleep problems, emotional distress, attachment styles with quality of life: comparison between gynaecological cancer survivors and noncancer women. J Clin Nurs 23(7-8):985–994. http://dx.doi.org/10.1111/jocn.12232

  • Author's Response 2

The inclusion criteria, sampling method, coding sequence, and assignment method of the research subjects are as follows:

This study was a randomized controlled trial that used a longitudinal design with two groups measured by a pretest and posttest. The inclusion criteria were females aged 20-75 years females with cancer who had undergone conventional cancer treatment for at least one month. ……... A total of 60 participants were referred by outpatient physicians according to the protocol. Then, according to the order of referral, the odd number participants were assigned to the MSM intervention group (n = 30), and the even number participants were assigned to the control group (n = 30).

  • Author's Response 3

We have added the illustration as follows.

The Chinese version of the FOR questionnaire was verified by Lee et al. (2018) [16] based on the Fear of Cancer Recurrence Inventory (FCRI) developed by Simard and Savard [17], with a total of 42 items. This measurement scale uses a five-point Likert scale. The degree of FOR in the past several months was measured. Participants who had higher scores had higher FOR symptoms. The exploratory factor analysis using Kaiser–Meyer–Olkin (KMO) measure (KMO value=0.915) and Bartlett’s Test of Sphericity (p<0.001). Fifteen items were retained and were divided into three domains: psychological distress, lifestyle function, and triggers; the Cronbach's α for each dimension was 0.956, 0.936, and 0.794, respectively. The overall Cronbach’s α of the internal consistency reliability of the scale was 0.954, indicating it was a reliable and practical measurement tool for assessing FOR.

  1. Lee YH, Lai GM, Lee DC, Tsai Lai LJ, Chang YP. (2018) Promoting physical and psychological rehabilitation activities and evaluating potential links among cancer-related fatigue, fear of recurrence, quality of life, and physiological indicators in cancer survivors. Integr Cancer Ther 17(4):1183-1194. https://doi.org/10.1177/1534735418805149
  • Author's Response 4

We have added the illustration as follows.

QOL was measured using the fourth Chinese version of the Functional Assessment of Cancer Therapy-General (FACT-G), authorized by the Assessment of Chronic Illness Therapy (FACIT) Institute, which was developed by Cella et al. (1993) [18], with a total of 27 items. This measurement scale uses a five-point Likert scale to measure QOL in the past week. This questionnaire was divided into four domains: physical well-being and emotional well-being dimensions, the higher the score is, the worse the QOL, and for the social/family well-being and functional well-being dimensions, the higher the score is, the better the QOL. The Cronbach’s α for the four subscales ranged between 0.81 and 0.93, and test-retest reliability ranged from 0.74 to 0.85, indicating excellent internal consistency [19].

  1. Cella DF, Tulsky DS, Gray G et al. (1993) The Functional Assessment of Cancer Therapy scale: development and validation of the general measure. J Clin Oncol 11(3):570-579. http://dx.doi.org/10.1200/JCO.1993.11.3.570
  • Author's Response 5

It was added in the fourth paragraph of the introduction section and the last paragraph of the discussion section.

Introduction:

However, malnutrition is widespread among cancer survivors, proper nutrition education can help them improve their quality of life [13], and complete nutritional supplementation can reduce the physical and mental stress caused by cancer [14].

Discussion:

One of the factors for this QOL improvement might have been the intake of a more balanced diet after attending the dietary training course. Good nutrition is an essential part of cancer treatment [13]. Sullivan et al., (2021) [24] indicated that most cancer survivors report serious nutritional problems after cancer, lacked correct dietary information, and hope to obtain more nutritional support to improve potential nutritional-related problems. Karim et al. (2021) [25] indicated that nutritional guidelines for cancer are associated with reduced cancer incidence and mortality, with breast cancer having the most considerable reduction, followed by endometrial cancer.

13.Byun M, Kim E, Kim J (2021) Physical and mental health factors associated with poor nutrition in elderly cancer survivors: insights from a nationwide survey. Int J Environ Res Public Health 18(17):9313. https://doi.org/10.3390/ijerph18179313

14. Borgi M, Collacchi B, Ortona E, Cirulli F (2020) Stress and coping in women with breast cancer: unravelling the mechanisms to improve resilience. Neurosci Biobehav Rev 119, 406-421. http://dx.doi.org/10.1016/j.neubiorev.2020.10.011

24. Sullivan ES, Rice N, Kingston E et al. (2021) A national survey of oncology survivors examining nutrition attitudes, problems and behaviours, and access to dietetic care throughout the cancer journey. Clin Nutr ESPEN 41:331–339. https://doi.org/10.1016/j.clnesp.2020.10.023

25. Karim S, Benn R, Carlson LE et al. (2021) Integrative oncology education: an emerging competency for oncology providers. Curr Oncol 28(1):853–862. https://doi.org/10.3389/10.3390/curroncol28010084

  • Author's Response 6

We revised.

This study still has research limitations. However, we distributed the instructional CDs of MSM courses and dietary cancer guidelines to the two groups of participants so that they could continue to practice at home. However, during the three-month follow-up period, a regular follow-up mechanism was not implemented to understand the practice conditions of the new lifestyle in daily life. Therefore, the long-term improvements in physical and mental health well-being may be compromised due to a lack of consistent practice or insufficient hours of training. In addition, there is no particular emphasis on stress-relieving diets in the two-hour nutrition session education, which is our future research direction.

  • Author's Response 7

It is not the purpose of our study to discuss the postoperative body changes and image reconstruction of breast cancer patients. Thank you for the valuable suggestions, which will be incorporated into future research.
